# Distributed Sequential Detection for Cooperative Spectrum Sensing in Cognitive Internet of Things

**DOI:** 10.3390/s24020688

**Published:** 2024-01-22

**Authors:** Jun Wu, Zhaoyang Qiu, Mingyuan Dai, Jianrong Bao, Xiaorong Xu, Weiwei Cao

**Affiliations:** 1School of Communication Engineering, Hangzhou Dianzi University, Hangzhou 310018, China; zhaoyangqiu@hdu.edu.cn (Z.Q.); dmy15168181146@163.com (M.D.); baojr@hdu.edu.cn (J.B.); xuxr@hdu.edu.cn (X.X.); 2National Mobile Communications Research Laboratory, Southeast University, Nanjing 211189, China; 3Key Laboratory of Flight Techniques and Flight Safety, CAAC, Civil Aviation Flight University of China, Guanghan 618307, China; ywcao@my.swjtu.edu.cn

**Keywords:** internet of thing, cooperative spectrum sensing, sequential detection rule, sensing time, cost function

## Abstract

The rapid development of wireless communication technology has led to an increasing number of internet of thing (IoT) devices, and the demand for spectrum for these devices and their related applications is also increasing. However, spectrum scarcity has become an increasingly serious problem. Therefore, we introduce a collaborative spectrum sensing (CSS) framework in this paper to identify available spectrum resources so that IoT devices can access them and, meanwhile, avoid causing harmful interference to the normal communication of the primary user (PU). However, in the process of sensing the PUs signal in IoT devices, the issue of sensing time and decision cost (the cost of determining whether the signal state of the PU is correct or incorrect) arises. To this end, we propose a distributed cognitive IoT model, which includes two IoT devices independently using sequential decision rules to detect the PU. On this basis, we define the sensing time and cost functions for IoT devices and formulate an average cost optimization problem in CSS. To solve this problem, we further regard the optimal sensing time problem as a finite horizon problem and solve the threshold of the optimal decision rule by person-by-person optimization (PBPO) methodology and dynamic programming. At last, numerical simulation results demonstrate the correctness of our proposal in terms of the global false alarm and miss detection probability, and it always achieves minimal average cost under various costs of each observation taken and thresholds.

## 1. Introduction

As wireless communication technology rapidly develops, spectrum resources cannot meet the growing number of internet of thing (IoT) devices and their applications. However, the frequency spectrum of primary users (PUs) still lies in an insufficient state in the time or space domain. To address this concern, cognitive radio (CR) is regarded as a prospective technology to identify available spectrum resources and allow IoT devices to opportunistically access them [1,2] without causing harmful interference to PUs [3]. But the spectrum-sensing behaviors of a single IoT device are susceptible to inherent factors of wireless propagation. Consequently, the cooperative spectrum sensing (CSS) paradigm is formulated to exploit spatial diversity and then improve the sensing accuracy of the PU signal through the observations of spatially positioning IoT devices. However, IoT architectures differ from traditional network architectures, which imply a high degree of reconfigurability, adaptability, mobility, and heterogeneity and present some insurmountable challenges to spectrum sensing. Traditional spectrum sensing techniques must be carefully redesigned for use in complex and scalable IoT systems [4].

In the past, some researchers have investigated spectrum sensing for IoT systems. An energy-efficient, reliable decision transmission in Zhu et al. to was proposed to decrease packet error and packet loss in industrial IoT [5]. In a low signal-to-noise ratio (SNR) environment, to minimize energy consumption and sensing time, Ansere et al. proposed a dynamic spectrum sensing algorithm [6]. Wan et al. proposed an energy-efficient CSS scheme to reduce the negative impact of spatial correlation [7]. Since the previous energy detector is usually limited by noise uncertainty, Miah et al. also proposed an energy-efficient CSS-based CR-enabled IoT network under the interference constraint [8]. Considering that battery-limited IoT devices are densely interconnected, Dao et al. optimized the sensing efficiency to leverage a lightweight but effective adaptive medium learning method [9]. Long et al. developed a harvesting-sensing-transmission tradeoff problem-based cognitive IoT to take the diversity of energy harvesting efficiency, spectrum sensing performance, and quality-of-service (QoS) of data transmission into consideration [10]. In order to enhance spectrum utilization in a 5G-based IoT, Abbas et al. proposed a hybrid mode of underlay and interweave-enabled scheme [11]. Gharib et al. proposed a heterogeneous multi-band multi-user CSS scheme to realize secondary users’ scheduling to sense a subset of channels in heterogeneous distributed CR networks [12]. Ejaz et al. presented a multiband CSS and resource allocation framework in a CR-enabled IoT 5G network to minimize energy consumption under the performance requirement [13]. To maximize the effective throughput, Zhang et al. jointly optimized the sensing time and packet error rate in cognitive IoT [14]. Miah et al. presented a CSS technique in a noise-uncertain environment to comprise the use of the Kullback–Leibler divergence in CR-based IoT [15]. To encourage spectrum sharing among unlicensed IoT devices, Lu et al. integrated the incentive mechanism into an orthogonal frequency division multiplexing (OFDM)-based cognitive IoT network with multiple unlicensed IoT devices in the context of incomplete information [16]. In the CSS of high real-time scenes of IoTs, Gao et al. considered an improved CSS scheme to decrease the latency and increase low throughput, where each cognitive node performs a truncated sequential probability ratio test (SPRT) over each observation vector [17]. Wu et al. achieved CSS between micro-sensing slots in cognitive unmanned aerial vehicle networks and approximated the error probability and the sensing time [18]. Moreover, an optimal CSS for CR networks is performed using offset quadrature amplitude modulation and universally filtered multicarrier non-orthogonal multiple access methodologies [19]. Mehmood et al. proposed an efficient QoS-based multi-path routing scheme for wireless body area networks [20]. In addition, Lin et al. investigated a destructive beamforming design in IoT networks from the perspective of a malicious active reconfigurable intelligent surface (RIS) and proposed a general optimization framework to solve the SNR minimization problem [21]. Ma et al. investigated the feasibility and performance of covert communication with a spectrum-sharing relay in the finite block length regime [22]. An et al. investigated the secrecy performance of a cognitive satellite-terrestrial network [23]. However, there is little research on the performance and efficiency of CSS for IoT devices.

A lot of efforts have been paid to CR-enabled IoT, considering issues such as achievable throughput, energy efficiency, frequency efficiency, or joint optimization with the spectrum resource allocation algorithm. These issues are also common on traditional CR networks. However, they did not take into account the cost issues in cognitive IoT, such as the sensing time and the cost of incorrect decisions, especially when considering CSS among multiple IoT devices. Hence, efficient spectrum sensing and resource allocation can be achieved only by realizing low-cost PU detection and ensuring spectrum sensing performance. Therefore, this article considers the optimal decision rule in cognitive IoT from the perspective of cost. To this end, a distributed cognitive IoT model is first established, including a pair of IoT devices for CSS and sequential detection, on the basis of which the sensing time and decision cost are defined, and the joint optimization problem between them is proposed. The optimal sensing time and threshold are analyzed by dynamic programming to obtain the optimal decision rule. The main contributions of this article can be summarized as follows:We formulate a distributed cognitive IoT model without a centralized fusion center (FC) and make use of energy detection to evaluate the local spectrum sensing performance of the IoT device. Furthermore, we also present a sequential detection framework to pave the way for the CSS of a pair of IoT devices.On the basis of the proposed distributed cognitive IoT and CSS models, the sensing time and decision cost are defined, and the joint optimization problem between them is proposed. Then, the person-by-person optimization (PBPO) approach is applied to distributed sequential detection to address this optimization problem.The optimal sensing time and threshold are analyzed by dynamic programming to obtain the optimal decision rule. At last, simulation results show the correctness and effectiveness of our proposed sequential detection rule in terms of sensing time and thresholds.

The remainder of this article is organized as follows: The local spectrum sensing model and sequential detection for CSS in a cognitive IoT are presented in Section 2. The optimal sensing time and decision rule based on distributed sequential detection are proposed and analyzed in Section 3. Comprehensive simulation result analyses and discussions are discussed in Section 4, and Section 5 draws a conclusion about this article.

## 2. System Model

### 2.1. Spectrum Sensing Model

In a cognitive IoT without a centralized FC, there is a PU and a pair of IoT devices participating in CSS, as shown in Figure 1. To protect the PUs normal operation from detrimental interference, each of the IoT devices $S_{1}$ and $S_{2}$ individually exploits spectrum sensing technology to sense the PU at the sensing slot and then derives a final local decision about the PUs presence through a predetermined combination rule through observations of the PU activity information at each multiple micro-sensing slot. According to the global decisions of IoT devices, a distributed CSS algorithm is adopted to derive a global decision after the sensing slot to decide whether to allow IoT devices to access the channel. At last, a pair of IoT devices are allowed to utilize the free spectrum band via a predetermined spectrum resource algorithm during the transmitting slot if the PU is declared as absent.

Energy detection is usually used as a local sensing technology because it is easy to implement and compatible with the PU network. In an energy detector, suppose that the hypotheses $H_{0}$ and $H_{1}$ represent the absence and presence of the PU, respectively, then the attenuated received PU signal at the k-th micro-sensing slot of an IoT device is expressed as [24]
(1)$$y_{k}\left(m\right)=\left\{\begin{array}{ll} n_{k}\left(m\right), & H_{0}\\ h_{k}s_{k}\left(m\right)+n_{k}\left(m\right), & H_{1} \end{array}\right.,$$
where m is the PU signal sampling, $n_{k}\left(m\right)$ is the circularly symmetric complex Gaussian (CSCG) noise, $s_{k}\left(m\right)$ is the complex-valued phase shift keying (PSK) signal at the PU, $n_{k}\left(m\right)$ and $s_{k}\left(m\right)$ are independent each other, $h_{k}$ is the channel gain. Then, the test static $E_{k}$ of energy detector is expressed by
(2)$$E_{k}=\frac{1}{M}\sum_{m=1}^{M}{\left|y_{k}\left(m\right)\right|}^{2},$$
where M is the sampling number of the received PU signal.

Following (2), we evaluate the local performance via a pre-determined detection threshold $\lambda_{k}$. Under the hypothesis $H_{0}$, the probability density function (PDF) $p_{0}\left(l\right)$ of the test static $E_{k}$ follows Chi-square distribution, the local false alarm probability is obtained by
(3)$$P_{f,k}=P\left(r_{k}=1|H_{0}\right)=P\left(E_{k}>\lambda_{k}|H_{0}\right)=\int_{\lambda_{k}}^{\infty }p_{0}\left(l\right)dl,$$
where $r_{k}$ is the sensing sample.

Suppose M is large enough, the PDF of $E_{k}$ is approximated as a Gaussian distribution where the mean $\mu_{0}=\sigma_{n}^{2}$, the variance $\sigma_{0}^{2}=\left[E{\left|n_{k}\left(m\right)\right|}^{4}-\sigma_{n}^{4}\right]/M$. Because $n_{k}\left(m\right)$ is CSCG, $E{\left|n_{k}\left(m\right)\right|}^{4}=2\sigma_{n}^{4}$, thus $\sigma_{0}^{2}=\sigma_{n}^{4}/M$. The sampling frequency is $f_{s}$, the duration time for the k-th micro-sensing slot is $\tau_{k}$, for simplicity of denotation, $M=\tau_{k}f_{s}$. Therefore, the local false alarm probability is given by
(4)$$P_{f,k}=Q\left(\left(\frac{\lambda_{k}}{\sigma_{n}^{2}}-1\right)\sqrt{\tau_{k}f_{s}}\right),$$
where
(5)$$Q\left(l\right)=\frac{1}{\sqrt{2\pi }}\int_{\lambda_{k}}^{\infty }exp\left(-\frac{t^{2}}{2}\right)dl.$$

Under the hypothesis $H_{1}$, PDF of $E_{k}$ is denoted by $p_{1}\left(l\right)$, the local detection probability can be expressed by
(6)$$P_{d,k}=P\left(r_{k}=1|H_{1}\right)=P\left(E_{k}>\lambda_{k}|H_{1}\right)=\int_{\lambda_{k}}^{\infty }p_{1}\left(l\right)dl.$$

Since the PDF of $E_{k}$ is also regarded as a Gaussian distribution where the mean $\mu_{1}=\left(1+\lambda_{k}\right)\sigma_{n}^{2}$, the variance $\sigma_{1}^{2}=\left[E{\left|h_{k}s_{k}\left(m\right)\right|}^{4}+E{\left|n_{k}\left(m\right)\right|}^{4}-{\left(h_{k}^{2}s_{k}^{2}\left(m\right)-\sigma_{n}^{2}\right)}^{2}\right]/M=\left[\left(1+\gamma_{k}\right)\sigma_{n}^{4}\right]/M$, where $\gamma_{k}$ is the received SNR at the k-th micro-sensing slot, the local detection probability can be given by
(7)$$P_{d,k}=Q\left(\left(\frac{\lambda_{k}}{\sigma_{n}^{2}}-\gamma_{k}-1\right)\sqrt{\frac{\tau_{k}f_{s}}{1+2\gamma_{k}}}\right).$$

### 2.2. Sequential Detection

Building on the above spectrum sensing model in a cognitive IoT, we further present a sequential detection framework for CSS and make the following assumptions and descriptions: The IoT device $S_{i}$ receives a sequence of observations $\left\{Z_{k}^{i}\right\}$, and $\left\{Z_{k}^{i}\right\}$ is i.i.d. and are independent of one another at a hypothesis, i=1,2. Under hypothesis $H_{j}$, the observations from the i-th IoT device follow a marginal probability density function $q_{j}^{\left(i\right)}$. In addition, the probability of hypotheses $H_{0}$ and $H_{1}$ are 1−ρ and ρ, respectively, a probability space is assumed to be $\left(\Omega ,F\right)=\left(R^{\infty }\times R^{\infty },B^{\infty }\times B^{\infty }\right)$ equipped with the probability measure $P=\rho P_{1}+\left(1-\rho \right)P_{0}$, where $P_{1}=P_{1}^{\left(1\right)}P_{1}^{\left(2\right)}$ and $P_{0}=P_{0}^{\left(1\right)}P_{0}^{\left(2\right)}$, $P_{j}^{\left(1\right)}$ and $P_{j}^{\left(2\right)}$ denote the restrictions of $P_{j}$ to the corresponding filtrations $\left\{F_{k}^{\left(i\right)}\right\}$ with $F_{k}^{\left(i\right)}=\sigma \left\{Z_{1}^{\left(i\right)},\ldots ,Z_{k}^{\left(i\right)}\right\}$. Each IoT device $S_{i}$ devises a sequential decision rule [25], $T^{\left(i\right)}$ is the time of stopping taking another sample, and $\theta^{\left(i\right)}$ takes the value 0 or 1 to declare whether one of two hypotheses is accepted.

## 3. Distributed Sequential Detection

According to the above model, we delve into the distributed sequential detection for a cognitive IoT in this section, including the optimal sensing time and the optimal sequential detection.

### 3.1. Problem Formulation

To study the cost problem of distributed sequential detection, we define a cost function $\Delta \left(\theta^{\left(1\right)},\theta^{\left(2\right)};H\right)$ to indicate the cost of error in any one or both of the decisions made by a pair of IoT devices. To be specific, $\Delta \left(0,\theta^{\left(2\right)};H_{1}\right)\geq \Delta \left(1,\theta^{\left(2\right)};H_{1}\right)$, $\Delta \left(1,\theta^{\left(2\right)};H_{0}\right)\geq \Delta \left(1,\theta^{\left(2\right)};H_{1}\right)$, $\Delta \left(1,\theta^{\left(2\right)};H_{0}\right)\geq \Delta \left(0,\theta^{\left(2\right)};H_{0}\right)$, and $\Delta \left(0,\theta^{\left(2\right)};H_{1}\right)\geq \Delta \left(0,\theta^{\left(2\right)};H_{0}\right)$. Similarly, the inequalities apply to $\theta^{\left(1\right)}$. From these inequalities, each additional sample of an IoT device also incurs a cost of c. Combining the time of stopping taking another sample and the cost function, there is a following decision problem, such as:(8)$$\underset{\left\{\left(T^{\left(i\right)},\theta^{\left(i\right)}\right)\right\}}{\inf}E\left\{{cT}^{\left(1\right)}+{cT}^{\left(2\right)}+\Delta \left(\theta^{\left(1\right)},\theta^{\left(2\right)};H\right)\right\}.$$

### 3.2. Preliminary Analysis

Since a positive cost c correlates with each additional time step taken by IoT devices in (8), the PBPO approach is applied to distributed sequential detection to address the problem of (8) [26]. Fixing $\left(T^{\left(2\right)},\theta^{\left(2\right)}\right)$, a stochastic optimization problem is described as
(9)$$J\left(\rho \right)=\underset{\left\{\left(T^{\left(1\right)},\delta^{\left(1\right)}\right)\right\}}{\inf}E\left\{{cT}^{\left(1\right)}+{cT}^{\left(2\right)}+\Delta \left(\theta^{\left(1\right)},\theta^{\left(2\right)};H\right)\right\}.$$

In (9), there is a special case, i.e., $\Delta \left(\theta^{\left(1\right)},\theta^{\left(2\right)};H\right)=\Delta \left(\theta^{\left(1\right)},H\right)+\Delta \left(\theta^{\left(2\right)},H\right)$, which is a classical sequential detection problem. Additionally, the cost function may be coupled between the two IoT devices.

Before solving (9), a sufficient statistic is preset as
(10)$$\rho_{k}^{\left(1\right)}=P\left(H=H_{1}|F_{k}^{\left(2\right)}\right),$$
and the recursion result from Bayes’ formula can be expressed as
(11)$$\rho_{k+1}^{\left(1\right)}=\frac{\rho_{k}^{\left(1\right)}q_{1}^{\left(1\right)}\left(x\right)}{\rho_{k}^{\left(1\right)}q_{1}^{\left(1\right)}\left(x\right)+\left(1-\rho_{k}^{\left(1\right)}\right)q_{0}^{\left(1\right)}\left(x\right)}P\left(H=H_{1}|F_{k}^{\left(2\right)}\right),$$
with $\rho_{0}^{\left(1\right)}=\rho$. Obviously, $\left\{\rho_{k}^{\left(1\right)}\right\}$ forms a Markov process about the filtration $\left\{F_{k}^{\left(1\right)}\right\}$.

Considering the finite horizon problem, the IoT device $S_{1}$ discontinues taking another sample and derives a decision not later than time τ. Let $J_{k}^{\tau }$ denote the minimal expected cost at the k-th micro-sensing slot, a dynamic programming equation.

(1)When $\left\{T^{\left(1\right)}=\tau \right\}$, we have


(12)
$$J_{T^{\left(1\right)}}^{\tau }\left(\rho_{T^{\left(1\right)}}^{\left(1\right)}\right)=inf\left\{E\left\{\Delta \left(0,\theta^{\left(2\right)};H\right)|F_{T^{\left(1\right)}}^{\left(1\right)}\right\},E\left\{\Delta \left(1,\theta^{\left(2\right)};H\right)|F_{T^{\left(1\right)}}^{\left(1\right)}\right\}\right\}.$$


(2)When $\left\{T^{\left(1\right)}=k\right\}$, k=1,…,τ−1, we have

(13)$$J_{T^{\left(1\right)}}^{\tau }\left(\rho_{T^{\left(1\right)}}^{\left(1\right)}\right)=inf\left\{E\left\{\Delta \left(0,\theta^{\left(2\right)};H\right)|F_{T^{\left(1\right)}}^{\left(1\right)}\right\},E\left\{\Delta \left(1,\theta^{\left(2\right)};H\right)|F_{T^{\left(1\right)}}^{\left(1\right)}\right\},c+\Delta_{k}^{\tau }\left(\rho_{k}^{\left(1\right)}\right)\right\},$$
where $\Delta_{k}^{\tau }\left(\rho_{k}^{\left(1\right)}\right)=E\left\{J_{k+1}^{\tau }\left(\rho_{k+1}^{\left(1\right)}\right)|F_{k}^{\left(1\right)}\right\}$.

Since $J_{0}^{\tau }$ is the minimal expected cost of the finite horizon problem, (12) and (13) provide the dependence of the minimal expected cost on the sufficient statistic $\rho_{k}^{\left(1\right)}$. It can be clearly seen from the right-hand side of unfolding (12), according to $E\left\{\Delta \left(0,\theta^{\left(2\right)};H\right)|F_{T^{\left(1\right)}}^{\left(1\right)}\right\}=\sum_{d=0}^{1}\sum_{j=0}^{1}P_{j}\left(\theta^{\left(2\right)}=d\right)\Delta \left(0,d;H_{j}\right)\times P\left(H=H_{j}|F_{T^{\left(1\right)}}^{\left(1\right)}\right)$, $E\left\{\Delta \left(1,\theta^{\left(2\right)};H\right)|F_{T^{\left(1\right)}}^{\left(1\right)}\right\}=\sum_{d=0}^{1}\sum_{j=0}^{1}P_{j}\left(\theta^{\left(2\right)}=d\right)\Delta \left(1,d;H_{j}\right)\times P\left(H=H_{j}|F_{T^{\left(1\right)}}^{\left(1\right)}\right)$, and using (8). The same holds true for (13), then we have $E\left\{J_{k+1}^{\tau }\left(\rho_{k+1}^{\left(1\right)}\right)|F_{k}^{\left(1\right)}\right\}=\int J_{k+1}^{\tau }\left(\rho_{k+1}^{\left(1\right)}\right)\left[\rho_{k}^{\left(1\right)}q_{1}^{\left(1\right)}\left(x\right)+\left(1-\rho_{k}^{\left(1\right)}\right)q_{0}^{\left(1\right)}\left(x\right)\right]dx$.

In addition, we define a function with respect to $\rho_{k}^{\left(1\right)}$ as $f\left(\rho_{k}^{\left(1\right)}\right)=min\left\{E\left\{\Delta \left(0,\theta^{\left(2\right)};H\right)|F_{T^{\left(1\right)}}^{\left(1\right)}\right\},E\left\{\Delta \left(1,\theta^{\left(2\right)};H\right)|F_{T^{\left(1\right)}}^{\left(1\right)}\right\}\right\}$, for all k=0, …τ, there are inequalities about $f\left(0\right)$ and $f\left(1\right)$ which follow their respective definitions, i.e.,
(14)$$f\left(0\right)<c+\Delta_{k}^{\tau }\left(0\right),$$
and
(15)$$f\left(1\right)<c+\Delta_{k}^{\tau }\left(0\right).$$

Moreover, the monotonicity results of $J_{k}^{\tau }\left(\rho \right)$ can be given by
(16)$$J_{k}^{\tau }\left(\rho \right)\leq J_{k+1}^{\tau }\left(\rho \right),\;0\leq \pi \leq 1,$$
and
(17)$$J_{k}^{\tau }\left(\rho \right)\leq \Delta_{k+1}^{\tau }\left(\rho \right),\;0\leq \pi \leq 1,$$
since each of the left-hand quantities is a hypo-mundum on a larger set of sensing times than the corresponding right-hand quantity.

### 3.3. Optimal Sensing Time

To solve problem (9), we consider the limit τ→∞, the pointwise limit of $J_{k}^{\tau }$ exists and is independent of k. More specifically, we have
(18)$$J\left(\rho \right)=\underset{\tau \to \infty }{\lim}J_{k}^{\tau }\left(\rho \right)=\underset{\tau \to \infty }{\lim}J_{k}^{\tau }\left(\rho \right),\;0\leq \rho \leq 1$$

Since CSS begins with two UAVs making decisions about the state of the PU signal within a defined sensing time, the collective sensing information from these UAVs leads to a comprehensive global decision regarding the state of the PU signal, adhering to a precise decision rule. To minimize decision costs, it is essential to determine the most opportune sensing time. To this end, we transform the initial problem (9) into (18), which aligns precisely with the dynamic programming equation (also known as Berman equation). In the dynamic programming equation, the long-term cost in a given sensing frame is equal to the cost from the current sensing time combined with the expected cost from the future actions taken at the following sensing time. Ultimately, we identify the optimal sensing time through this dynamic programming equation by following Theorem 1.

**Theorem 1.** *The minimal expected cost on *$J\left(\rho \right)$ *satisfies the dynamic programming equation [27]*

(19)$$J\left(\rho \right)=min\left\{E\left\{\Delta \left(0,\theta^{\left(2\right)};H\right)\right\},E\left\{\Delta \left(1,\theta^{\left(2\right)};H\right)\right\},c+\Delta_{J}\left(\rho \right)\right\},\;0\leq \rho \leq 1,$$*where* $\Delta_{J}\left(\rho \right)=E\left\{J\left(\rho_{1}\right)\right\},\;0\leq \rho \leq 1$.

*The optimal sensing time is*(20)$$T_{opt}=inf\left\{k|\rho_{k}^{\left(1\right)}\notin \left(\xi_{L}^{\left(1\right)},\xi_{U}^{\left(1\right)}\right)\right\},$$ *where a pair of thresholds* 
$\left(\xi_{L}^{\left(1\right)},\xi_{U}^{\left(1\right)}\right)$ *are described as*
(21)$$\xi_{L}^{\left(1\right)}=sup\left\{0\leq \rho \leq \frac{1}{2}|c+\Delta_{J}\left(\rho \right)=E\left\{\Delta \left(0,\theta^{\left(2\right)};H\right)\right\}\right\},$$
*and*
(22)$$\xi_{U}^{\left(1\right)}=inf\left\{1/2\leq \rho \leq 1|c+\Delta_{J}\left(\rho \right)=E\left\{\Delta \left(1,\theta^{\left(2\right)};H\right)\right\}\right\},$$

**Proof of Theorem 1.** Taking the limit of (13) and using (18) and (19) follows. The concavity of J derives from the limit of concave functions. Inequalities like (14) and (15) also hold. Utilizing these inequalities, the concavity of $\Delta_{J}$, and $J\left(\rho \right)$, the optimal sensing time is the threshold type, as shown in (20), where the threshold is determined by(23)$$c+\Delta_{J}\left(\xi_{L}^{\left(1\right)}\right)=E\left\{\Delta \left(0,\theta^{\left(2\right)};H\right)\right\}{|}_{\left\{\rho =\rho_{L}^{\left(1\right)}\right\}},$$
and
(24)$$c+\Delta_{J}\left(\xi_{U}^{\left(1\right)}\right)=E\left\{\Delta \left(1,\theta^{\left(2\right)};H\right)\right\}{|}_{\left\{\rho =\rho_{U}^{\left(1\right)}\right\}}$$This establishes the proposition. □

### 3.4. Optimal Decision Rule

Similar to an argument used in the proof of Proposition 7.4 [25], the uniqueness of the limit value function for (9) follows. Moreover, since the optimal thresholds $\xi_{L}^{\left(1\right)}$ and $\xi_{U}^{\left(1\right)}$ are coupled from (14) and (15), two simultaneous dynamic programming equations should be solved.

Given a value of $\Delta \left(T^{\left(2\right)},\theta^{\left(2\right)}\right)$, the optimal local decision rule of the IoT device $S_{1}$ is derived, and vice versa. That is to say, when two IoT devices achieve their respective optimal decisions for each other’s optimal decision rule, as a result, the global optimal decision rules can be iteratively implemented by continuously fixing the threshold of one IoT device and optimizing the threshold of the other by Theorem 1.

Finally, there are following processes at the optimal decision rule of the IoT devices $S_{i}$, i=1, 2, such as, (1) if $\rho_{k}^{\left(i\right)}\leq \xi_{L}^{\left(i\right)}$, the decision rule accepts $H_{0}$; (2) if $\rho_{k}^{\left(i\right)}\geq \xi_{U}^{\left(i\right)}$, the decision rule accepts accept $H_{1}$; (3) if $\xi_{L}^{\left(i\right)}\leq \rho_{k}^{\left(i\right)}\leq \xi_{U}^{\left(i\right)}$, the decision rule continues taking another sample, where a pair of thresholds $\left(\xi_{L}^{\left(i\right)},\xi_{U}^{\left(i\right)}\right)$ at the per-IoT device are obtained by
(25)$$\xi_{L}^{\left(i\right)}=\frac{{\bar{P}}_{m}^{\left(i\right)}}{1-{\bar{P}}_{f}^{\left(i\right)}},$$
and
(26)$$\xi_{U}^{\left(i\right)}=\frac{1-{\bar{P}}_{m}^{\left(i\right)}}{{\bar{P}}_{f}^{\left(i\right)}},$$
where ${\bar{P}}_{m}^{\left(i\right)}$ and ${\bar{P}}_{f}^{\left(i\right)}$ are the tolerable miss detection probability and the tolerable false alarm probability, respectively.

A similar method can be utilized for the quickest detection problem. In such a problem, each of the IoT devices $S_{j}$ sequentially receives observations $\left\{Z_{k}^{\left(j\right)}\right\}$, then there exists a change point t following a geometric distribution with a mass at 0, and correspondingly there is a known marginal density $q_{0}^{\left(j\right)}$ for k=1, …, t−1 and $q_{1}^{\left(j\right)}$ for k=t, …. Given the change point, IoT device observations are assumed to be conditionally independent, and they are valid within IoT devices and across IoT devices. Now, in order to quickly detect the change point and control the false alarm probability, each IoT device needs to optimally select sensing times $T^{\left(i\right)}$ (each measurable with respect to their own filtrations $F^{\left(i\right)}$) with the aim of minimizing $E\left\{\Delta \left(T^{\left(1\right)},T^{\left(2\right)};\;t\right)\right\}$, where $\Delta \left(T^{\left(1\right)},T^{\left(2\right)};\;t\right)=1_{\left\{T^{\left(1\right)}<t\right\}}1_{\left\{T^{\left(2\right)}<t\right\}}+c_{1}\left(T^{\left(1\right)}-t\right)1_{\left\{T^{\left(1\right)}\geq t\right\}}++c_{2}\left(T^{\left(2\right)}-t\right)1_{\left\{T^{\left(2\right)}\geq t\right\}}$. Therefore, the optimal solution can be given by
(27)$$T^{\left(1\right)}=inf\left\{k|P\left(t\leq k|F_{k}^{\left(1\right)}\right)\geq \xi_{1}^{*}\right\},$$
and
(28)$$T^{\left(2\right)}=inf\left\{k|P\left(t\leq k|F_{k}^{\left(2\right)}\right)\geq \xi_{2}^{*}\right\},$$
where a pair of optimal thresholds $\xi_{1}^{*}$ and $\xi_{2}^{*}$ are coupled via a system of two dynamic programming equations. The term $1_{\left\{T^{\left(1\right)}<t\right\}}1_{\left\{T^{\left(2\right)}<t\right\}}$ appears in the cost function that couples the solution.

## 4. Simulation Results

In this section, simulation results are introduced to corroborate the correctness and effectiveness of our proposal with respect to the global performance and the average cost of an IoT device. To this end, in 10^6^ spectrum sensing frames, unless otherwise specified, some parameter settings are considered as follows: The number of micro-sensing slots is 20, the probability ρ of the hypothesis $H_{1}$ is 0.5, and the local detection probability and the local false alarm probability are set to be 0.6 and 0.4, respectively. Both the tolerable false alarm probability and the tolerable false alarm probability vary from 0.01 to 0.3 within an interval of 0.01.

### 4.1. Performance Analysis

Figure 2 illustrates the relationship between the global false alarm probability $Q_{f}$ and the tolerable false alarm probability ${\bar{P}}_{f}$ under various tolerable miss detection probabilities. First of all, it can be seen that as the tolerable false alarm probability becomes more relaxed, the global false alarm probability shows a stepwise increase, and the larger the tolerable false alarm probability, the larger the gradient of the step. This is because for a fixed probability, an increase in the tolerable false alarm probability leads to a decrease in the upper threshold $\xi_{U}$, and the sequential detection rule is easier to accept $H_{1}$, which in turn results in an increase in the global false alarm probability. Meanwhile, it is worth noting that on the steps before the global false alarm probability jumps, although the tolerable false alarm probability continues to increase, the global false alarm probability remains unchanged. At this point, an increase in the initial sensing time does not bring about a change in the global false alarm probability; that is, an increase in observation does not bring about a change in the global false alarm probability, and the initial sensing time is the optimal sensing time.

Moreover, the impact of the tolerable miss detection probability on the global false alarm probability can be neglected at the beginning. That is to say, the thresholds $\left(\xi_{L},\xi_{U}\right)$ of the sequential detection rule are still not satisfied. But as the tolerable false alarm probability increases, the impact of the tolerable missed detection probability becomes more and more obvious. To be specific, the larger the tolerable miss detection probability, the faster the global false alarm probability jumps. Apparently, the larger the tolerable miss detection probability, the larger the upper threshold $\xi_{U}$, resulting in a more acceptable $H_{1}$.

Under various tolerable miss detection probabilities, the relationship between the global miss detection probability $Q_{m}$ and the tolerable false alarm probability ${\bar{P}}_{f}$ is shown in Figure 3. In contrast to Figure 2, the tolerable false alarm probability has a greater effect on the global miss detection probability than the global false alarm probability, and the effect is positive. In details, when the tolerable false alarm probability increases from 0.01 to 0.3, correspondingly, the global miss detection probability basically goes down from 0.95 to 0.22. Since the lower threshold $\xi_{L}$ increases as the tolerable false alarm probability increases according to (25), the sequential detection rule is prone to accept $H_{0}$, resulting in a decrease in the global miss detection probability. Furthermore, in such an environment, the global miss detection probability of a large tolerable miss detection probability decreases first because it increases the lower threshold $\xi_{L}$, i.e., ${\bar{P}}_{m}=0.2$.

In addition, similar to Figure 2, the steps before the global miss detection probability jumps indicate that although the tolerable false alarm probability continues to increase, the global false alarm probability remains unchanged. At this point, an increase in the initial sensing time does not bring about a change in the global miss detection probability; that is, an increase in observation does not bring about a change in the global miss detection probability, and the initial sensing time is the optimal sensing time.

Next, we further take the impact of the tolerable miss detection probability on global performance given a fixed tolerable false alarm probability into consideration. As displayed in Figure 4, regardless of the tolerable miss detection probability, it is obvious that a large tolerable false alarm probability leads to a low upper threshold $\xi_{U}$, therefore being prone to accept $H_{1}$. However, it also should be noted that as the tolerable miss detection probability increases, the global false alarm probability under different tolerable false alarms has jitter at different positions, such as jitter up at ${\bar{P}}_{f}=0.05,\;0.1,\;02$ and jitter down when ${\bar{P}}_{f}=0.2$. This is not a surprise and is a direct result of the fact that a pair of tolerable probabilities simultaneously change and the decision condition is reached within a certain sensing time.

Similar to the global miss detection probability in Figure 4, given the tolerable false alarm probability, the positive impact of the tolerable miss detection probability is illustrated in Figure 5. In particular, the trend of the global miss detection probability is exactly opposite to that of the global false alarm probability, and the change interval is larger. There is no doubt that the tolerable miss detection probability makes the lower threshold $\xi_{L}$ smaller so that $H_{1}$ is easier to accept.

Following the joint impact of the tolerable performance metrics on the global performance, we further simulate the optimal cost of the tolerable performance under various costs of each observation taken, where the cost of each observation taken c is set to be 0.1 and 1. As shown in Figure 6, for a pair of fixed-tolerable performances, the larger the cost of each observation taken, c, the larger the average cost. Moreover, as the tolerable false alarm probability increases, the average cost decreases. This is to say, an increasing tolerable false alarm probability makes the lower/upper threshold larger/smaller, resulting in the global decision being difficult to make. Consequently, the sensing time increases. However, the increasing tolerable false alarm probability also makes the global miss detection probability decrease, as shown in Figure 3. As a result, the global miss detection probability dominates the average cost because the cost of miss detection decreases.

Similar to Figure 6, the higher cost of each observation in Figure 7 leads to a higher average cost. In line with the global miss detection probability depicted in Figure 5, the average cost follows. The simulation result also confirms once again that global missed detection dominates the average cost. In summary, following the PBPO methodology, the optimal sequential detection rule can be reached as the sensing environments to minimize the cost at an IoT device.

### 4.2. Performance Comparision

Following the proposed sequential detection and the tolerable miss detection probability ${\bar{P}}_{m}=0.3$, the performance comparison with classical sequential detection and Neyman-Pearson (N-P) is further promoted in the following Figure 8. Firstly, we can see that both the false alarm and miss detection probabilities of N-P are basically not affected by the increase in tolerable false alarm probability, and they are basically fixed at 0.42 and 0.18, respectively. This is because N-P is a fixed-sample-size rule (with the tolerable false alarm probability as a single threshold), and the required sensing time is also fixed. When taking the optimal sensing time of the proposed sequential detection rule as the stopping time of N-P, its performance will not be affected by a single threshold because the tolerable false alarm probability as a threshold may be too large or too small. Secondly, compared with N-P, two sequential detection rules are affected by both tolerable false alarm and miss detection probabilities. Figure 8 shows that the tolerable false alarm probability gradually increases, the global false alarm probability gradually increases, and the missed detection probability gradually decreases in a stepwise manner. However, after optimization, the sequential detection rule, whether in terms of false alarm probability or missed detection probability, has less performance than the classic one. That is to say, under the same sensing time (the optimal sensing time), the sequential detection performance proposed in this paper is the best. Finally, it should be noted that as the tolerable false alarm probability increases, that is, the probability of deciding that the PU is present also increases, so the global false alarm probability also increases. At the same time, the false alarm probability decreases. It can also be seen that there is a trade-off between false alarm and miss detection probabilities.

In addition to the above results, other rules of the same type (such as Bayesian detection) cannot meet the performance comparison under the influence of the miss detection probability with a fixed false alarm probability. In addition, the comparison of average cost can also be seen from the performance of N-P, because the only difference between Bayesian detection and N-P is the threshold. Therefore, under the same local decision cost, the average cost of N-P is still higher than that of the proposed sequential detection rule.

## 5. Conclusions

In this article, we delved into the distributed sequential detection rule for CSS in the context of cognitive IoTs. To begin, we established a spectrum sensing model within the periodic spectrum sensing frame structure and presented a sequential detection framework. Based on this framework, we defined the sensing time and cost functions for IoT devices and formulated an optimization problem regarding average cost. Furthermore, we employed the PBPO method to solve this finite horizon problem, enabling us to analyze the optimal sensing time for optimal sequential detection. Finally, through a series of numerical simulations, we demonstrated the accuracy and effectiveness of our proposed sequential detection rule in terms of sensing time and thresholds.

## Figures and Tables

**Figure 1 sensors-24-00688-f001:**
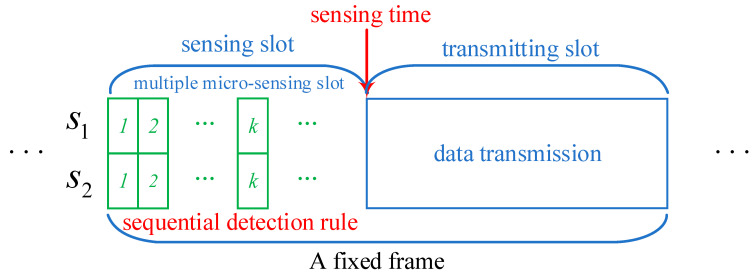
The periodic spectrum sensing frame structure of a cognitive IoT.

**Figure 2 sensors-24-00688-f002:**
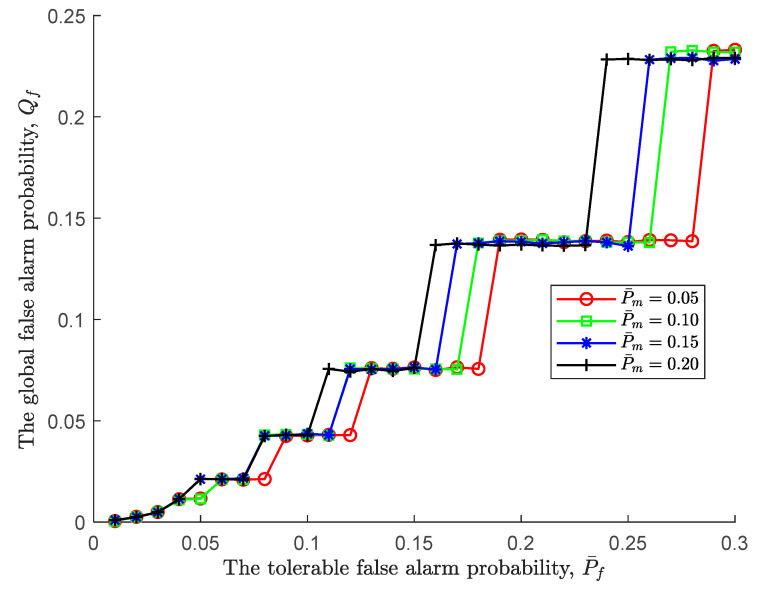
The global false alarm probability vs. the tolerable false alarm probability.

**Figure 3 sensors-24-00688-f003:**
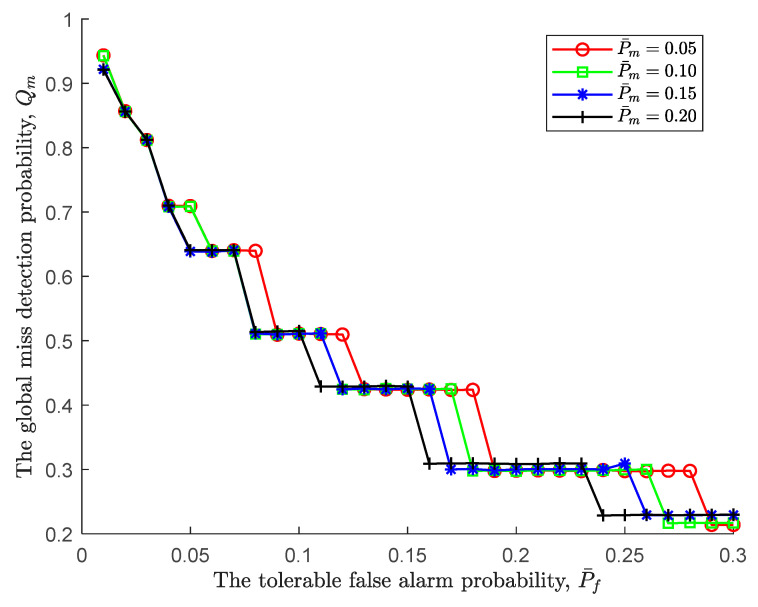
The global miss detection probability vs. the tolerable false alarm probability.

**Figure 4 sensors-24-00688-f004:**
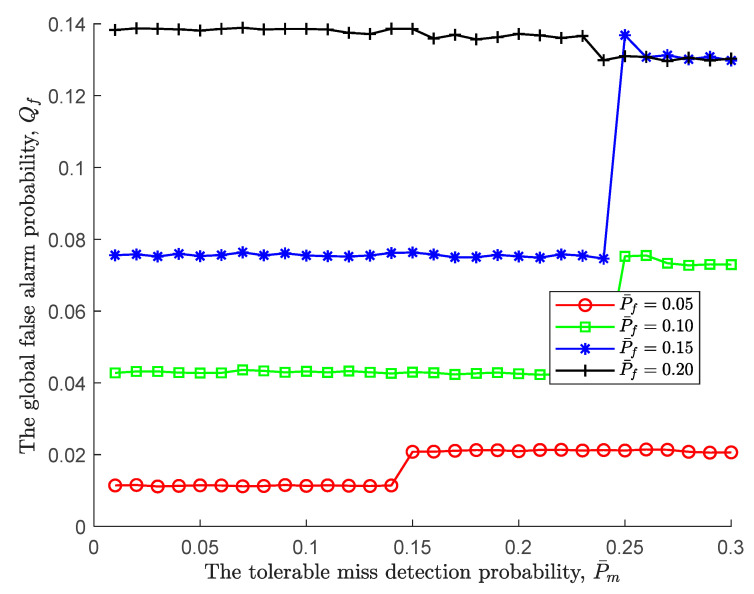
The global false alarm probability vs. the tolerable miss detection probability.

**Figure 5 sensors-24-00688-f005:**
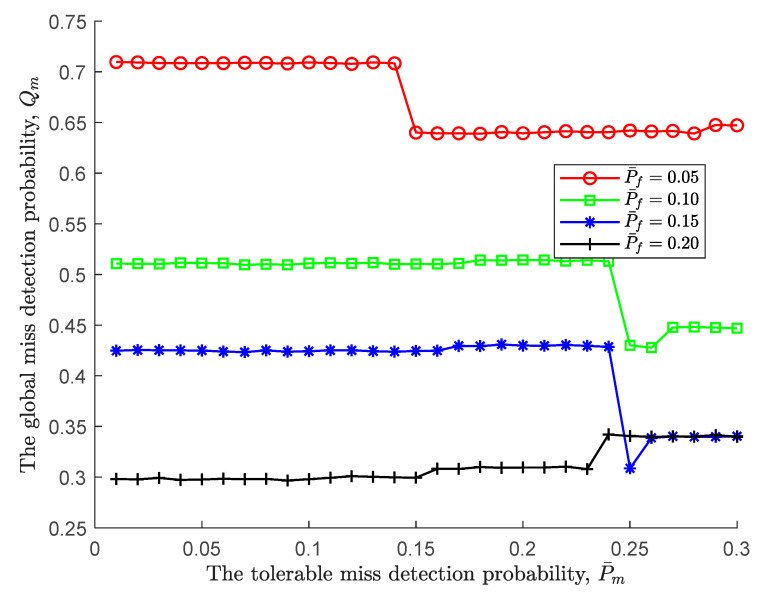
The global miss detection probability vs. the tolerable miss detection probability.

**Figure 6 sensors-24-00688-f006:**
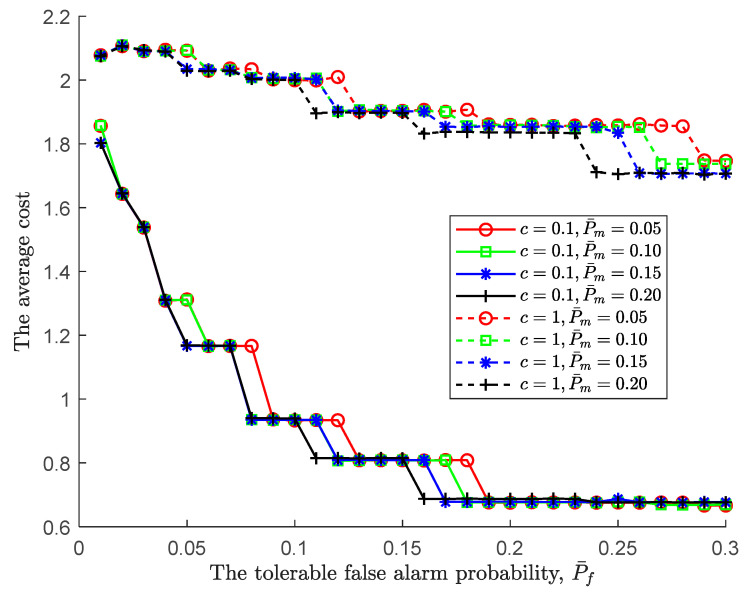
The average cost vs. the tolerable false alarm probability under various costs of each observation taken.

**Figure 7 sensors-24-00688-f007:**
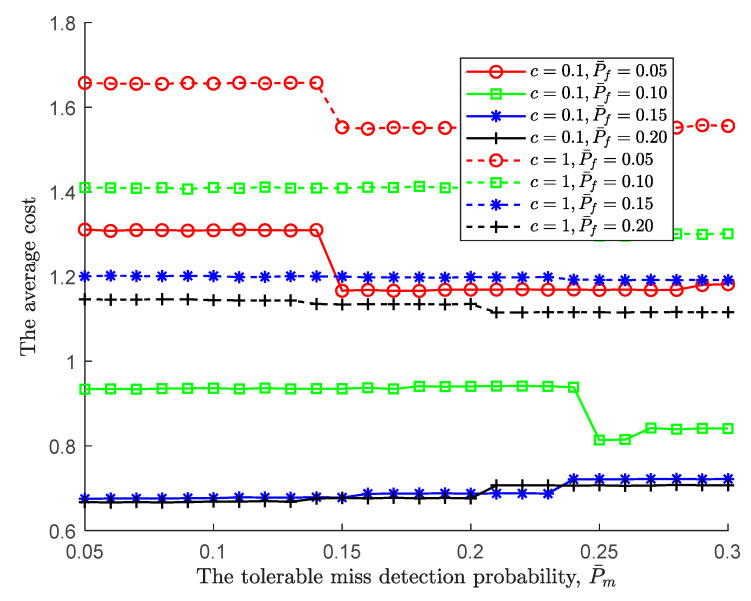
The average cost vs. the tolerable miss detection probability under various costs of each observation taken.

**Figure 8 sensors-24-00688-f008:**
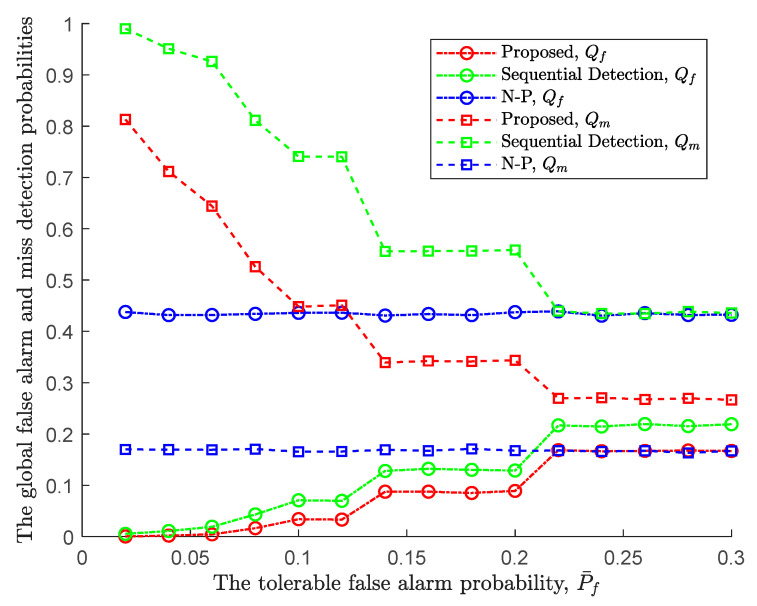
The global false alarm and miss detection probabilities of three rules vs. the tolerable false alarm probability.

## Data Availability

Data are contained within the article.

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
