# Peer review of "Distributed Sequential Detection for Cooperative Spectrum Sensing in Cognitive Internet of Things"

_sensors, 2024, doi:10.3390/s24020688_

Round 1
Reviewer 1 Report
Comments and Suggestions for Authors
Please see the attached file.

Comments on the Quality of English LanguageMinor editing of English language is needed.
Author Response
Comments and Suggestions for Authors
This paper introduced a collaborative spectrum sensing framework to identify available spectrum resources, so that IoT devices can access it and meanwhile avoid causing harmful interference to the normal communication of the PU. Generally, the authors seem have done a solid work, however, some parts need to be further clarified.
The reviewer has the following concerns:
1. In abstract, it is suggested to briefly and clearly introduce the background and facing challenges, the first problem about the available spectrum resources detection is clear, while the second problem of “stopping time and decision cost” would confuse readers. The authors could use more popular sentences to demonstrate it.
Reply: Thanks for your review. In abstract, we have changed the origin text to “The rapid development of wireless communication technology has led to the increasing number of internet of thing (IoT) devices, and the demand for spectrum for these devices and their related applications is also increasing. However, spectrum scarcity has become an increasingly serious problem. Therefore, ….”.
Additionally, it should be noted that since we use the stopping theory to solve the problem of cooperative spectrum sensing, we have introduced the concept of stopping time. In fact, for cooperative spectrum sensing, stopping time is the sensing time. But in order to avoid ambiguity, we have uniformly changed the stopping time to sensing time. The cost of determining the signal status of the PU in several situations that IoT devices incur, such as the cost of correctly determining the PU status, which we consider to be 0, and the cost of incorrectly determining the PU status. In order for readers to have a clearer understanding of this concept, we have further explained the decision in the abstract, i.e., “… the issue about the sensing time and decision cost (the cost of determining whether the signal state of the PU is correct or incorrect) arises.” |
As the full name and abbreviation of IoT appear at the introduction, which also need to be added in abstract.
Reply: Thanks for your review. We have revised and added in line 15, i.e., “…The rapid development of wireless communication technology has led to the increasing number of internet of thing (IoT) devices, and the demand for spectrum for these devices and their related applications is also increasing. However, spectrum scarcity has become an increasingly serious problem…” |
At the first glance of the key word “stopping time”, the reviewer is quite confusing before reading the main context and figure 1. It is suggested to replace this term with “sensing slot ratio” or other words, which would make the meaning more clear, if the term “stopping time” is not an official and well-recognized one.
Reply: Thanks for your review. According to comments, we have uniformly changed the stopping time to sensing time. Please refer to the response in Q1 for details. |
The contributions need to be further clarified. The authors should explain the difference of this paper with the existing ones on sensing slot optimization. Moreover, the novelty and contributions of this paper also need to be emphasized, instead of only introducing what has been done in this paper.
Reply: Thanks for your review. According to comments, we have added the part of contributions in paragraph 4-6 in the revised manuscript, i.e., “The main contributions of this article can be summarized as follows • We formulate a distributed cognitive IoT model without a centralized fusion center (FC), and make use of the energy detection to evaluate the local spectrum sensing performance for the IoT device. Furthermore, we also present a sequential detection framework to pave the way for CSS of a pair of IoT devices. • On the basis of the proposed distributed cognitive IoT and CSS model, the sensing time and decision cost are defined, and the joint optimization problem between them is proposed. Then, the person-by-person optimization (PBPO) approach is applied to distributed sequential detection to address this optimization problem. • The optimal sensing time and threshold are analyzed by dynamic programming to obtain the optimal decision rule. At last, simulation results show that correctness and effectiveness of our proposed sequential detection rule in terms of the sensing time and thresholds.” |
In the introduction, it is suggested to introduce the following recent works in IoT and cognitive radio fields to highlight the state-of-art of this paper:
[R1] “Pain without gain: Destructive beamforming from a malicious RIS perspective in IoT networks,” IEEE Internet of Things Journal, early access, Sep. 2023, DOI: 10.1109/JIOT.2023.3316830.
[R3] “Covert communication with a spectrum sharing relay in the finite blocklength regime,” China Communications, vol. 20, no. 4, pp. 195-211, Apr. 2023.
[R4] “Secure transmission in cognitive satellite terrestrial networks,” IEEE Journal on Selected Areas in Communications, vol. 34, no. 11, pp. 3025-3037, Nov. 2016.
Reply: Thanks for your review. We have cited the following three references in Refs. [21]-[23] of the revised manuscript, i.e., “In addition, Lin et al. investigated a destructive beamforming design in IoT networks from the perspective of a malicious active reconfigurable intelligent surface (RIS), and proposed a general optimization framework to solve the SNR minimization problem [21]. Ma et al. investigated the feasibility and performance of the covert communication with a spectrum sharing relay in the finite blocklength regime [22]. An et al. investigated the secrecy performance of a cognitive satellite terrestrial network [23].” |
The authors claimed that “this article considers the optimal decision rule”, the optimality needs to be proved and explained.
Reply: Thanks for your review. The details of the optimal decision rule can be found in Section 3.4. |
The comparison to existing works in simulations is limited.
Reply: Thanks for your review. According to the review comments, we have added subsection 4.2, such as: 4.2. Performance Comparision Following the proposed sequential detection and the tolerable miss detection probability , the performance comparison with classical sequential detection and Neyman-Pearson (N-P) is further promoted in the following Figure 8. Firstly, we can see that both the false alarm and miss detection probabilities of N-P are basically not affected by the increase of tolerable false alarm probability, and they are basically fixed at 0.42 and 0.18, respectively. This is because N-P is a fixed-sample-size rule (with the tolerable false alarm probability as a single threshold), and the required sensing time is also fixed. When taking the optimal sensing time of the proposed sequential detection rule as the stopping time of N-P, its performance will not be affected by a single threshold, because the tolerable false alarm probability as a threshold may be too large or too small. Secondly, compared with N-P, two sequential detection rules are affected by both tolerable false alarm and miss detection probabilities. Figure 8 shows that the tolerable false alarm probability gradually increases, and the global false alarm probability gradually increases while the miss detection probability gradually decreases in a stepwise manner. However, after optimization, the sequential detection rule, whether in terms of false alarm probability or missed detection probability, has less performance than the classic one. That is to say, under the same sensing time (the optimal sensing time), the sequential detection performance proposed in this paper is the best. Finally, it should be noted that as the tolerable false alarm probability increases, that is, the probability of deciding that the PU is present also increases, so the global false alarm probability also increases. At the same time, the false alarm probability decreases. It can also be seen that there is a trade-off between false alarm and miss detection probabilities.
Figure 8. The global false alarm and miss detection probabilities of three rules vs the tolerable false alarm probability. In addition to the above results, other rules of the same type (such as Bayesian detection) cannot meet the performance comparison under the influence of the miss detection probability with fixed false alarm probability. In addition, the comparison of average cost can also be seen from the performance of N-P, because the only difference between Bayesian detection and N-P is the threshold. Therefore, under the same local decision cost, the average cost of N-P is still higher than that of the proposed sequential detection rule. |
The authors should carefully check the whole paper to avoid the writing and grammar errors, such as additional “)” in line 80, unlogic sentence “while ensuring spectrum sensing performance can efficient spectrum sensing and resource allocation be achieved”, and “Most of these efforts are focused on”.
Reply: Thanks for your review. We have deleted the extra “)”, and changed “while ensuring spectrum sensing performance can efficient spectrum sensing and resource allocation be achieved” to “Hence, efficient spectrum sensing and resource allocation can be achieved only by realizing low-cost PU detection and ensuring spectrum sensing performance”, “Most of these efforts are focused on” to “A lot of efforts have been paid to”. Additionally, we have conducted detailed inspections and proofreading in other areas, but we will not point them out one by one here. |

Reviewer 2 Report
Comments and Suggestions for Authors
I have gone through the article titled Distributed Sequential Detection for Cooperative Spectrum Sensing in Cognitive Internet of Things. The overall structure and representation of the article are good, but still, some edits are required to increase the appeal and longevity of the article. 1. The abstract does not correlate with the content of the article. Please rewrite the abstract and try to make it more informative.2. The lines in the conclusion section are not well synchronized and interlinked; please rewrite them.3. The article must be checked for linguistic and grammatical errors.4. The proof of cooperative spectrum sensing is not clear. The authors should elaborate on the proof so that it is easily understandable for readers.5. There are many typos and inconsistencies in the writing.6. The problem definition is explained clearly, but a still more relevant explanation is expected that the author could refer to and cite recent papers, i.e.,
a. Balachander, T., Ramana, K., Mohana, R. M., Srivastava, G., & Gadekallu, T. R. (2023). Cooperative Spectrum Sensing Deployment for Cognitive Radio Networks for Internet of Things 5G Wireless Communication. Tsinghua Science and Technology, 29(3), 698-720.
b. Mehmood, G., Khan, M. Z., Bashir, A. K., Al-Otaibi, Y. D., & Khan, S. (2023). An Efficient QoS-Based Multi-Path Routing Scheme for Smart Healthcare Monitoring in Wireless Body Area Networks. Computers and Electrical Engineering, 109, 108517.
7. All the references must be uniform and according to the journal format.
Comments on the Quality of English Language
Minor typo mistakes need correction.
Author Response
Comments and Suggestions for Authors
I have gone through the article titled Distributed Sequential Detection for Cooperative Spectrum Sensing in Cognitive Internet of Things. The overall structure and representation of the article are good, but still, some edits are required to increase the appeal and longevity of the article.
1. The abstract does not correlate with the content of the article. Please rewrite the abstract and try to make it more informative.
Reply: We have rewritten the abstract section in order to provide readers with a better understanding of the content and core of our research at a glance, i.e., “The rapid development of wireless communication technology has led to the increasing number of internet of thing (IoT) devices, and the demand for spectrum for these devices and their related applications is also increasing. However, spectrum scarcity has become an increasingly serious problem. Therefore, we introduce a collaborative spectrum sensing (CSS) framework in this paper to identify available spectrum resources, so that IoT devices can access it and meanwhile avoid causing harmful interference to the normal communication of the primary user (PU). However, in the process of sensing the PU’s signal in IoT devices, the issue about the sensing time and decision cost (the cost of determining whether the signal state of the PU is correct or incorrect) arises. To this end, we propose a distributed cognitive IoT model, which includes two IoT devices independently using sequential decision rules to detect the PU. On this basis, we define the sensing time and cost function for IoT devices, and formulate an average cost optimization problem in CSS. To solve this problem, we further regard the optimal sensing time problem as a finite horizon problem, and solve the threshold of the optimal decision rule by person-by-person optimization (PBPO) methodology and dynamic programming. At last, numerical simulation results demonstrate the correctness of our proposal in terms of the global false alarm and miss detection probability, and it always achieves minimal average cost under various cost of each observation taken and thresholds.” |
The lines in the conclusion section are not well synchronized and interlinked; please rewrite them.
Reply: Thanks for your review. We have made revisions to the conclusion section as follows, “In this article, we delved into the distributed sequential detection rule for CSS in context of cognitive IoTs. To begin, we established a spectrum sensing model within the periodic spectrum sensing frame structure and presented a sequential detection framework. Based on this framework, we defined the sensing time and cost function for IoT devices and formulated an optimization problem regarding average cost. Furthermore, we employed the PBPO method to solve this finite horizon problem, enabling us to analyze the optimal sensing time for the optimal sequential detection. Finally, through a series of numerical simulations, we demon-started the accuracy and effectiveness of our proposed sequential detection rule in terms of sensing time and thresholds.” |
The article must be checked for linguistic and grammatical errors.
Reply: Thanks for your review. We have changed “while ensuring spectrum sensing performance can efficient spectrum sensing and resource allocation be achieved” to “Hence, efficient spectrum sensing and resource allocation can be achieved only by realizing low-cost PU detection and ensuring spectrum sensing performance”, “Most of these efforts are focused on” to “A lot of efforts have been paid to”. Additionally, we have conducted detailed inspections and proofreading in other areas, but we will not point them out one by one here. |
The proof of cooperative spectrum sensing is not clear. The authors should elaborate on the proof so that it is easily understandable for readers.
Reply: Thanks for your review. The proof of cooperative spectrum sensing mentioned by the reviewer is actually closely related to the optimal sensing time, which is also the basis and starting point for us to finally find the optimal decision rule.
Firstly, cooperative spectrum sensing actually involves two UAVs making decision on the state of the primary user signal within a certain sensing time, and then the sensing information from the two UAVs makes the final global decision (on the state of the primary user signal) through a specific decision rule. In order to minimize the decision cost during the decision process, we need to determine the optimal sensing time. To this end, we will transform the initial problem (9) into problem (18), which exactly satisfies the Berman equation (a.k.a. dynamic programming equation) which tells us what long-term reward can we expect, given the state we are in and assuming that we take the best possible action now and at each subsequent step. In the end, we found the optimal sensing time through this dynamic programming equation.
In order to help readers better understand this point, we have provided a detailed explanation after formula (18), such as “Since CSS begins with two UAVs making decisions about the state of the PU signal within a defined sensing time, the collective sensing information from these UAVs leads to a comprehensive global decision regarding the state of the PU signal, adhering to a precise a decision rule. To minimize decision cost, it's essential to determine the most opportune sensing time. To this end, we transform the initial problem (9) into (18), which aligns precisely with the dynamic programming equation (a.k.a. Berman equation). In the dynamic programming equation, the long-term cost in a given sensing frame is equal to the cost from the current sensing time combined with the expected cost from the future actions taken at the following sensing time). Ultimately, we identify the optimal sensing duration through this dynamic programming equation through following Theorem 1.” |
There are many typos and inconsistencies in the writing.
Reply: Thanks for your review. We have deleted the extra “)” in the original manuscript. Additionally, we have conducted detailed revisions in other text, but we will not point them out one by one here. |
The problem definition is explained clearly, but a still more relevant explanation is expected that the author could refer to and cite recent papers, i.e.,- Balachander, T., Ramana, K., Mohana, R. M., Srivastava, G., & Gadekallu, T. R. (2023). Cooperative Spectrum Sensing Deployment for Cognitive Radio Networks for Internet of Things 5G Wireless Communication. Tsinghua Science and Technology, 29(3), 698-720.
- Mehmood, G., Khan, M. Z., Bashir, A. K., Al-Otaibi, Y. D., & Khan, S. (2023). An Efficient QoS-Based Multi-Path Routing Scheme for Smart Healthcare Monitoring in Wireless Body Area Networks. Computers and Electrical Engineering, 109, 108517.
Reply: Thanks for your review. We have cited the following three references in Refs. [19]-[20] of the revised manuscript, i.e., “Moreover, an optimal CSS for CR networks is performed using Offset quadrature amplitude modulation universal filtered multicarrier non-orthogonal multiple access methodologies [19]. Mehmood et al. proposed an efficient QoS-based multi-path routing scheme for wireless body area networks [20].” |
- All the references must be uniform and according to the journal format.
Reply: Thanks for your review. We have modified the previous reference format, and the newly cited references are strictly formatted according to the journal's reference format. |

Reviewer 3 Report
Comments and Suggestions for Authors
Paper seems suitable for publication in current form
Author Response
Comments and Suggestions for Authors
Paper seems suitable for publication in current form.
Reply: Thanks for your review. |

Reviewer 4 Report
Comments and Suggestions for Authors
The author proposes a sequential detection framework for CSS in a cognitive IoT.
The issue addressed in the paper is well-defined.
A more elaborate related work should be included in the paper to demonstrate the importance of the results obtained.
In section 4, it is specified that simulations were carried out, but no information is presented as to how they were carried out. These informations are necessary to be able to reproduce these simulations. Also, a comparison with other solutions from the literature should be presented.
What are the weak points and strong points of the proposed solution?
Author Response
Comments and Suggestions for Authors
The author proposes a sequential detection framework for CSS in a cognitive IoT.
Reply: Thanks for your review. |
The issue addressed in the paper is well-defined.
Reply: Thanks for your review. |
A more elaborate related work should be included in the paper to demonstrate the importance of the results obtained.
Reply: Thanks for your review. On one hand, we further added our contributions of this article in the introduction section to demonstrate the importance of the results obtained, i.e., “The main contributions of this article can be summarized as follows • We formulate a distributed cognitive IoT model without a centralized fusion center (FC), and make use of the energy detection to evaluate the local spectrum sensing performance for the IoT device. Furthermore, we also present a sequential detection framework to pave the way for CSS of a pair of IoT devices. • On the basis of the proposed distributed cognitive IoT and CSS model, the sensing time and decision cost are defined, and the joint optimization problem between them is proposed. Then, the person-by-person optimization (PBPO) approach is applied to distributed sequential detection to address this optimization problem. • The optimal sensing time and threshold are analyzed by dynamic programming to obtain the optimal decision rule. At last, simulation results show that correctness and effectiveness of our proposed sequential detection rule in terms of the sensing time and thresholds.”
On the other hand, we also add more simulation results to demonstrate the importance of the results obtained, please refer to the following Q4. |
In section 4, it is specified that simulations were carried out, but no information is presented as to how they were carried out. These information are necessary to be able to reproduce these simulations. Also, a comparison with other solutions from the literature should be presented.
Reply: Thanks for your review. According to the review comments, we have added subsection 4.2, such as: 4.2. Performance Comparision Following the proposed sequential detection and the tolerable miss detection probability , the performance comparison with classical sequential detection and Neyman-Pearson (N-P) is further promoted in the following Figure 8. Firstly, we can see that both the false alarm and miss detection probabilities of N-P are basically not affected by the increase of tolerable false alarm probability, and they are basically fixed at 0.42 and 0.18, respectively. This is because N-P is a fixed-sample-size rule (with the tolerable false alarm probability as a single threshold), and the required sensing time is also fixed. When taking the optimal sensing time of the proposed sequential detection rule as the stopping time of N-P, its performance will not be affected by a single threshold, because the tolerable false alarm probability as a threshold may be too large or too small. Secondly, compared with N-P, two sequential detection rules are affected by both tolerable false alarm and miss detection probabilities. Figure 8 shows that the tolerable false alarm probability gradually increases, and the global false alarm probability gradually increases while the miss detection probability gradually decreases in a stepwise manner. However, after optimization, the sequential detection rule, whether in terms of false alarm probability or missed detection probability, has less performance than the classic one. That is to say, under the same sensing time (the optimal sensing time), the sequential detection performance proposed in this paper is the best. Finally, it should be noted that as the tolerable false alarm probability increases, that is, the probability of deciding that the PU is present also increases, so the global false alarm probability also increases. At the same time, the false alarm probability decreases. It can also be seen that there is a trade-off between false alarm and miss detection probabilities.
Figure 8. The global false alarm and miss detection probabilities of three rules vs the tolerable false alarm probability. In addition to the above results, other rules of the same type (such as Bayesian detection) cannot meet the performance comparison under the influence of the miss detection probability with fixed false alarm probability. In addition, the comparison of average cost can also be seen from the performance of N-P, because the only difference between Bayesian detection and N-P is the threshold. Therefore, under the same local decision cost, the average cost of N-P is still higher than that of the proposed sequential detection rule. |
What are the weak points and strong points of the proposed solution?
Reply: Thanks for your review. The weakness is that as the number of UAVs increases, the computational complexity of the proposed method will become higher, which is not conducive to deriving the optimal sequential decision rule. The advantage is that most traditional data fusion methods only consider centralized cooperative spectrum sensing, which means that the perception results of multiple drones are reported to the fusion center through a reporting channel, and then the fusion center makes a global decision on the presence or absence of the primary user signal. This results in lower efficiency for centralized cooperative spectrum sensing, while distributed cooperative spectrum sensing does not need to consider this issue, Moreover, combining sequential decision-making can also improve the efficiency of collaborative spectrum sensing. Considering the flexible location of UAVs, distributed systems are more suitable for dynamic network topologies. |

Round 2
Reviewer 1 Report
Comments and Suggestions for Authors
The authors have well addressed all my concerns, no further comments.
Reviewer 2 Report
Comments and Suggestions for Authors
The author addressed my concerns so I accept the paper in it current form.